# Sustainability of Global Economic Policy and Stock Market Returns in Indonesia

Shabir Mohsin Hashmi [1], Muhammad Akram Gilal [2] and Wing-Keung Wong [3,4,5,*]

1   School of Economics and Management, Yancheng Institute of Technology, Yancheng 224051, China; hashmishabbir@163.com
2   Department of Economics, University of Sindh, Jamshoro 76080, Pakistan; akram.gilal@usindh.edu.pk
3   Department of Finance, Fintech Center and Big Data Research Center, Asia University, Taichung City 41354, Taiwan
4   Department of Medical Research, China Medical University, Taichung City 40447, Taiwan
5   Departments of Economics and Finance, The Hang Seng University of Hong Kong, Siu Lek Yuen 41354, Hong Kong, China
*   Correspondence: wong@asia.edu.tw

**Abstract:** Interdependence in trade and financial globalization has increased the vulnerability of developed and developing countries to external shocks alike, whereas emerging markets are more vulnerable to the shocks originating from the world's leading economies. This paper investigates the impact of the uncertainty from the global economic policy on the return of the Indonesian stock market by using the time-varying correlation based on the rolling window method and time-varying built dynamic conditional correlation method. Both the rolling window and condition correlation estimates indicate that the correlation between global policy uncertainty and Indonesian stock returns is time-varying. The results of the autoregressive distributed lag-based regression indicate that inflation, global crude oil prices, gross domestic product, and world crude oil production have significant impacts on the dynamic conditional correlation. The average negative estimate of time-varying correlation suggests that investors when faced with liquidity constraints in one country may sell off their assets in another country to raise funds in order to meet their future financial needs. This also indicates that the rise in the uncertainty of economic policy in developed markets has a negative impact on the shocks faced by the Indonesian stock market. Based on our empirical findings, it is recommended that Indonesian policymakers should place more focus on the sustainability of the economic growth, pay close attention to volatile crude oil prices, world crude oil production, and inflation so as to avoid dynamic interaction between the uncertainty of economic policy in the developed markets and the return of the Indonesian stock market.

**Keywords:** stock market returns; spill-over effect; dynamic conditional correlation; economic policy uncertainty

**JEL Classification:** C10; F30; F41; F43; G15

## 1. Introduction

Indonesia is the largest economy in South East Asia [1], and it is considered an emerging economy. However, Indonesian financial institutions and equity markets are distinctive from other emerging stock markets and financial institutions. Indonesia registered 7.9 percent growth from 2008 to 2017; such growth is higher than many major emerging economies, including those of Brazil, Russia, India, China, South Africa, Singapore, and Malaysia. In addition, Islamic features differentiate the Indonesian equity market from stock markets in other emerging countries [2]. Moreover, the integration of the Indonesian equity market with global equity markets has increased over time. The country allowed foreign investors to buy 49 percent of new and listed shares, except bank shares, as of

September 1989 [3]. In an effort to liberalize its equity market, in July 1992, the Indonesian government ceased its control of the Jakarta Stock Exchange (JSX) and vested it as a private limited company regulated by member companies [4]. As a result, foreign institutions' share of total market capitalization reached 41 percent in 2007. Furthermore, Indonesia's sustainable economic growth in the post-East Asian crisis period has further been driven by its exports [5]. However, liberalization of trade and financial integration with the rest of the world has increased Indonesia's vulnerability to external shocks, particularly shocks hitting the world's large economies. This is apparent from the recent global financial crisis that hit the U.S. economy in 2008–2009,which caused a 30 percent depreciation of Indonesian currency, IDR, against the USD and a 51.1 percent decrease in the country's stock market prices in 2008 [6]. Hence, it is appropriate to identify linkages between the uncertainty of economic policy in the developed markets and the return of the Indonesian stock market.

A nonzero probability of changes in existing economic policies that determine the rules of the game for economic agents is called economic policy uncertainty [7]. It is countercyclical and affects almost the entire economy. A recent surge in research interest in the effects of policy uncertainty has been due to the global financial crisis, quantification of uncertainty, and rise in computing power [8]. It has primarily focused on the effects of uncertainty on a large set of macroeconomic variables, such as inflation and unemployment [9], firm-level investment [10,11], demand for durable goods [12] and output growth [13,14]. The effect of uncertainty on investment is more severe for firms with larger irreversible investments and government spending dependency [15]. Economic policy uncertainty with firm-level uncertainty leads to economic contraction [9,14,16].

Stock markets play a key role in the economic growth of a country by improving liquidity, mobilizing capital, exercising corporate control, risk pooling, and sharing services including investment levels. Stock market development augments economic growth by attracting more investment [17]. Increased financial globalization has led to greater integration of equity markets around the world [18]. This is apparent from a large number of studies that examine comovements among stock markets. Ref. [19] attributes fluctuation in most of the stock market in the sample countries to a common global factor and equates it with international stock market comovements. Ref. [20] examines the correlation among thirty-two emerging equity markets from four regions and find its significant presence within regions, across regions, and comovements with the rest of the world's equity markets. Ref. [21] identifies a volatility spillover across the stock markets of London, New York, and Tokyo. The U.S. is the largest equity market in the world. Due to its size, a large number of studies have focused on the U.S. equity market spillover effect. Ref. [22] examines day-to-day linkages between the U.S. and four Asian equity market prices and finds a significant tendency among the Asian markets to follow U.S. equity market prices. Ref. [23] shows a significant interaction between U.S. stock market uncertainty and emerging market returns. Ref. [24] indicates that the U.S. equity market has a significant effect on the equity market returns of France, Germany, and the United Kingdom. According to [25], the U.S. equity market skewness negatively predicts international equity market returns. Ref. [26] identifies significant mean return and volatility spillover effects from U.S. equity markets on the stock markets of Brazil, Russia, India, China, and South Africa. Ref. [27] indicates risk spillover between G7 and U.S. equity markets.

Ref. [28] identifies uncertainty as a new channel for financial market contagion and stock market volatility. Government policies set the environment in which private businesses have to operate. Both real and financial markets react negatively to government-policy-related uncertainty [7,29]. An increase in uncertainty hampers long-run growth prospectus and equity prices [30]. The U.S. constitutes 23.89 percent of the global economy [31] and more than 40 percent of the world equity market. Hence, shocks hitting the U.S not only affect the U.S. economy but also spread to other countries. Ref. [32] argues that trade-related fluctuations in the world's largest economy (U.S.) influence financial markets around the world. Ref. [33] identifies a negative effect of the tightening of monetary policy in the U.S. on 50 equity markets around the globe. The recent most global financial crisis

that hit the U.S. financial markets also spread to other equity markets around the globe. It resulted in 27, 51, and 55 percent falls in the U.S., Europe, and Japan stock market prices, respectively. The fall in emerging market equity prices was quite large and stood at 60 percent [34].

This paper makes at least three contributions to the empirical literature on economic policy uncertainty and stock market returns. First, there are many studies in the literature examining the interaction between economic policy uncertainty and emerging economies' equity markets, including those of Brazil, China, India, South Korea, Malaysia, Mexico, Philippines, Russia, South Africa, Taiwan, and Thailand; see, for example, [35–44]. However, as far as we know, there is no previous study examining the interaction between economic policy uncertainty and the equity market for Indonesia. Thus, the first contribution of our paper is that we bridge the gap in the literature regarding the examination of the interaction between economic policy uncertainty and the equity market in Indonesia. The second contribution of our paper to the empirical literature on economic policy uncertainty and stock market returns is that our paper is the first paper employing both rolling window correlation [45] and the dynamic conditional correlation method [46] to examine the interaction between the Indonesian equity market and the uncertainty of economic policy in the developed markets.

Another contribution of our paper is that we observe the factors that determine the dynamic interaction between the uncertainty of economic policy in the developed markets and the return of the Indonesian stock market. Ref. [47] identifies increased comovements in the international equity market during the recession. According to [48], uncertainty effects are intensified during the recession in the U.S. economy. Ref. [39] finds a negative effect of policy uncertainty on G7 stock markets in the bearish regime and a significant negative role in the bearish and bullish market of BRIC. Ref. [38] find a negative effect of uncertainty on stock market returns during bearish regimes for both developed and emerging economies but in higher magnitude for the latter countries. Using quantile regression, ref. [49] finds an increase in the out-of-sample predictability of economic policy uncertainty when stock market performance is poor to moderate. According to [50], the link between policy uncertainty and equity returns increases during poor economic conditions. This means that bad economic conditions could increase the vulnerability of the stock market returns to uncertainty shock. In this paper, we expand on the above work to determine the factors that explain the time-varying-based dynamic conditional correlation between the uncertainty of economic policy in the developed markets and the return of the Indonesian stock market. Our study is based on two hypotheses: the first hypotheses ($H_{01}$) assumes that the correlation between policy uncertainty in the global economy and return of the Indonesian market is constant; the second hypothesis ($H_{02}$) tests whether oil price shocks, macroeconomic variables, and recessionary indicators affect the dynamic conditional correlations between global economic policy uncertainty and Indonesian stock market return.

The remainder of the paper is organized as follows: A literature review is given in Section 2 followed by a discussion on data and methodology in Section 3. This includes data discussion in Section 3.1 followed by a discussion on rolling window correlation, dynamic conditional correlation, and autoregressive distributed lag model in Sections 3.2.1–3.2.3, respectively. The empirical analysis is provided in Section 4, and it includes descriptive statistics, rolling window correlation, and dynamic conditional correlation; the results of the autoregressive distributed lag model are presented in Sections 4.1–4.4. The conclusion is given in Section 5.

## 2. Literature Review

There is a vast body of literature linking economic policy uncertainty and stock returns. Refs. [51–53], and others examine the detrimental economic effects of monetary, fiscal, and regulatory policy uncertainty. However, the bulk of the literature is largely built on four channels through which policy uncertainty affects asset prices. First, firms and other

economic agents may alter their consumption and savings decisions due to uncertainty. As a result, consumers increase their precautionary savings, which potentially reduces consumption expenditure [30]. On the other hand, a high level of uncertainty impels firms to freeze prospective investment projects and hiring [10]. Second, the uncertainty effect of demand and supply might lead to a rise in production cost and financing [54]. Policy uncertainty not only reduces levels of investment, hiring, and consumption but also hampers economic growth, particularly in smaller but open economies. Third, to counter the negative effect of policy uncertainty, governments could adopt a protectionist policy that may further increase risk in financial markets [29]. Lastly, due to policy uncertainty, a decrease in future cash flows or an increase in the risk-adjusted discount rate or both may affect stock prices [55].

Several studies have been conducted on policy uncertainty, although among all, two approaches are noteworthy. The first approach is event based with respect to the date of policy implementation. Despite being well documented, an event-based approach may be artificially precise [56]. The second approach uses government elections as a proxy for policy uncertainty [57,58]. The study conducted by [59] revealed that the equity market remains volatile a month before the presidential elections. Ref. [60] also came up with a similar conclusion that political uncertainty leads to greater market volatility but after, not before, major unexpected political outcomes such as Brexit. As is well known, when the economy is doing well, politicians stick with their old policies, but when the economy is not stirring in the right directions, politicians are tempted to experiment, consequently spurring further uncertainty [29].

Ref. [61] finds a negative correlation between U.S. economic policy uncertainty and returns of all high-yielding currencies, except JPY. Ref. [7] identifies the key role of uncertainty in terms structure dynamics, which, in turn, has a major effect on countercyclical volatility of asset returns.

The effect of the economic policy uncertainty index constructed by [7] on economic activities is much prevalent among the work of researchers [8,14]. Since its development, many studies follow similar techniques for evaluating the effects of policy uncertainty on different macroeconomic indicators, stock market returns, and stock market volatility.

In recent years, numerous studies have further been added to the subject, although the work of [28,39–41,43,44,49,56,62–77] is prominent. All of these studies relate their own country's policy uncertainty to their own country's stock market returns, with the exception of a few. In this regard, refs. [28,40,43,44,64,69,70,72,76,77] are some of the exceptions. Ref. [43] examines the effect of U.S. economic policy uncertainty on G7 and IBSA (India, Brazil, and South Africa) countries' stock markets. Ref. [64] relates their own country's economic policy uncertainty and global uncertainty (policy uncertainty in China, the European area, Japan, and the USA) to the stock market returns of Hong Kong, Malaysia, and South Korea. Results based on causality in quantile regression provide strong evidence of relevancy of one's own country's economic policy uncertainty and global uncertainty for Malaysian stock market returns and South Korean stock market returns and their volatility. Hong Kong stock returns appear unaffected by both kinds of uncertainties. Ref. [39] also employs the quantile regression techniques to examine the dependence structure between economic policy uncertainty and stock market returns with respect to G7 and BRIC countries. Ref. [44] examines their own country's policy uncertainty and the effects of U.S. economic policy uncertainty on the stock market returns of Pacific-Rim countries (Australia, Canada, China, Japan, Korea, and the U.S.). Pooled vector autoregression results suggest a negative effect of policy uncertainty on the returns of all stock markets. However, the U.S. policy uncertainty appears an insignificant determinant of Australian stock market returns. Ref. [40] examines volatility spillovers between U.S. economic policy uncertainty and BRIC (Brazil, Russia, India, and China) equity markets. Ref. [70] focuses on the effect of U.S. economic policy uncertainty on China's A/B stock markets and the U.S. stock market returns co-movement. Ref. [69] also examines the spillover effect of U.S. economic policy uncertainty on global financial markets with respect

to nineteen economies. Factor augmented vector autoregression results showed a negative spillover effect of U.S. economic policy uncertainty on all countries except the Chinese equity market. Ref. [72] examines the importance of European and U.S. economic policy uncertainty in predicting European stock market returns represented by the stock markets of the UK, Germany, and France. The results indicate the failure of their own country's economic policy uncertainty in improving forecast accuracy of these markets. On the other hand, U.S. economic policy uncertainty offers valuable information that enhanced the predictability of these stock market returns. Ref. [28] investigates the effect of U.S. economic policy uncertainty, financial uncertainty, and news-implied uncertainty on the stock market volatility of six industrialized and three emerging economies. Economic policy uncertainty and news-implied uncertainty have both positive and negative effects on the returns of stock markets. Ref. [76] identifies a negative effect of U.S. economic policy uncertainty on BRIC countries' stock market returns, except for those of China. Ref. [77] shows that the U.S. instead of China's economic policy uncertainty has a key role in shaping global financial markets.

In a recent publication, ref. [78] examines the impact of changes in economic policy uncertainty on the Japanese stock return. The results indicate that a rise in volatility creates a negative effect on stock prices, reaffirming the risk premium hypothesis. It is further stated that the degree of asymmetry of uncertainty on stock returns is significant for the Japanese market as compared with the U.S. influence. Ref. [79] also identifies a significant volatility spillover effect from the U.S. equity market to stock markets in the southeast Asian countries.

The aforementioned studies conclude a mixture of positive and negative effects of policy uncertainty on stock returns. We further extend the discussion and attempt to pinpoint those factors that determine the dynamic linkages between the uncertainty of economic policy in the developed markets and the return of the Indonesian stock market.

## 3. Data and Methodology

### 3.1. Data

In this study, we employed multiple sources of data. Our sample period ranges from January 2000 to December 2017. The data series used in this study are the global economic policy uncertainty (GEPU), Jakarta stock market returns, consumer price index, global crude oil prices, gross domestic product, world crude oil production, and dummy variable representing the recessionary indicator. The uncertainty of economic policy in the developed markets is a GDP-weighted average of national economic policy uncertainty indices of 21 countries. These countries account for 71 percent and 80 percent of global output on purchasing power parity basis and market exchange rates, respectively. Data on Jakarta stock market returns (*JKSE*) were obtained from Bloomberg and are defined as $[\log(JKSE_t) - (\log(JKSE_{t-1})]$.

Regarding the world crude oil production and world crude oil prices, we obtained data from the US Energy Information Administration, while data on the consumer price index, gross domestic product, and a recessionary indicator were taken from the database of the Organization of Economic Cooperation and Development (OECD). In the absence of access to industrial production index data, monthly real GDP was obtained by interpolating the quarterly nominal GDP adjusted with the consumer price index.

*RI* represents the recessionary period and equals one when the economy is in recession and zero otherwise. The period beginning at the midpoint of a peak and ending at the midpoint of a trough is called a recession. Hence, the entire period of the peak and trough is included in the recessionary indicator. OECD defines three recessionary periods during the sample period: $RI_1$ (2003M3 to 2004M4), $RI_2$ (May 2008 to June 2009), and $RI_3$ (February 2013 to December 2016).

### 3.2. Methodology

The existing empirical literature provides evidence of unstable and time-varying linkages between policy uncertainty and stock market returns. In this study, our prime objectives are to examine the time-varying linkages between the variables and to examine what factors potentially determine the correlation. To comprehend, how the correlation between different time series evolves over time, we employed both-rolling window correlation and time-varying-based dynamic conditional correlation [46]. A combination of both techniques is rarely found in contemporary literature.

### 3.2.1. Rolling Window Correlation

The rolling window correlation is a powerful tool to estimate a correlation between two time series. There are two main advantages of rolling window correlation [62,80]. First, it allows correlation among the variables to vary over time. Second, it enables us to distinguish between times of positive and negative correlation and sub-samples of high correlation during the sample period [40].

In this paper, the rolling window of fifty observations is used to estimate the rolling correlation between the variables. The first rolling window of a fixed length of fifty observations gives the first rolling correlation coefficient. The sample is then rolled over to calculate the second correlation coefficient for the second window and so forth. We then drop the first observation and use an observation ranging from month 2 to month 52 to calculate the correlation for the second window. This process is repeated until the last window counts the last fifty observations.

### 3.2.2. Dynamic Conditional Correlation

The rolling window is a good estimate, but, fundamentally, it suffers from some inherent weaknesses. First, it uses subsample information. Second, results are sensitive to window size. Third rolling window correlation cannot estimate time-varying correlations properly when the relationship between the variables is unstable [80]. To overcome the limitations of the rolling window correlation, we applied [46] dynamic conditional correlation (DCC hereafter) to estimate the time-varying correlation between the return of the Indonesian stock market and the uncertainty of economic policy in the developed markets. The advantages of using the DCC approach are that it is not necessary to set up any window size, there is no loss of observation, and there is no requirement for any subsample estimation [62].

Furthermore, DCC allows time variation in the conditional correlation to avoid the curse of dimensionality of multivariate generalized autoregressive heteroscedasticity (MGARCH) models. This is achieved by separately specifying the conditional volatilities and the conditional correlations. The latter are then modeled in terms of a small number of parameters to avoid the dimensionality problem of MGARCH models [81].

Assume $y_t = [y_{1t}, y_{2t}]'$ is a $2 \times 1$ vector that contains the data series whose conditional mean equation in the reduced form VAR can be written as

$$A(L)y_t = \varepsilon_t, \ \varepsilon_t \sim \ (0, \ H_t), \ t = 1 \ldots .T \tag{1}$$

where $A$ is a matrix, $L$ is the lag operator, and $\varepsilon_t$ is a vector of innovation thathas the following conditional covariance matrix:

$$H_t = D_t R_t D_t \tag{2}$$

while $D_t = \text{diag}\{\sqrt{h_{i,t}}\}$. Here, $h_{i,t}$ is a $2 \times 2$ matrix, including the time-varying standard deviations estimated from the generalized autoregressive conditional heteroscedasticity (GAHRCH $(p, q)$ model, and $R_t = \rho_{i,j,t} = \frac{q_{i,j,t}}{\sqrt{q_{ii,t}q_{jj,t}}}$ is the $2 \times 2$ matrix containing dynamic conditional correlation, which is the main focus of this paper.

### 3.2.3. Autoregressive Distributed Lag Model

An autoregressive distributed lag (ARDL) model introduced by [82] includes one or more lagged values of the endogenous variable and current and lagged values of one or more exogenous variables. In this paper, we use the following simple ARDL model to identify which factors can explain the dynamic conditional correlation:

$$DCC_t = \alpha + \beta_1 \Delta cpi_t + \beta_2 \Delta gcop_t + \beta_3 \Delta gdp_t + \beta_4 \Delta wcop_t + \beta_5 RI_t + \varepsilon_t \tag{3}$$

where $\alpha$ is constant; $\Delta cpi_t$, $\Delta gcop_t$, $\Delta gdp_t$, and $\Delta wcop_t$ represent the consumer price index, global crude oil prices, real income, and world crude oil production, respectively; $\Delta$ represents the first difference operator, $\varepsilon_t$ represents stochastic disturbance, and $RI_t$ is the dummy variable that equals 1 during a recession as defined by the Organization of Economic Co-operation and Development and zero otherwise.

The ARDL version of Equation (3) is:

$$\Delta DCC_t = + \sum_{i=1}^{p} \beta_{1i} DCC_{t-i} + \sum_{i=1}^{p} \beta_{2i} \Delta cpi_{t-i} + \sum_{i=1}^{p} \beta_{3i} \Delta gcop_{t-i} + \sum_{i=1}^{p} \beta_{4i} \Delta gdp_{t-i} + \sum_{i=1}^{p} \beta_{5i} \Delta wcop_{t-i}$$

$$\beta_6 RI + \beta_7 DCC_{t-1} + \beta_8 cpi_{t-1} + \beta_9 gcop_{t-1} + \beta_{10} gdp_{t-1} + \beta_{11} wcop_{t-1} + \beta_{12} ect_{t-1} + \varepsilon_t \tag{4}$$

where $\alpha$ and $\varepsilon$ are the intercept and error terms, respectively; $\beta_1$ to $\beta_6$ represent the short term estimates; $\beta_8$ to $\beta_{11}$ are the long-run estimates; and $\beta_{12}$ represents the speed of adjustment, and its estimate must be negative and significant to validate the presence of a long-run relationship among the variables. The null hypothesis of no long-run relationship among the variables is tested by the following hypotheses:

$$H_0 : \beta_8 = \beta_9 = \beta_{10} = \beta_{11} = 0 \text{ against the alternative hypothesis}$$
$$H_a : \beta_8 \neq \beta_9 \neq \beta_{10} \neq \beta_{11} \neq 0$$

As an improvement of Johansen cointegration test [83], ARDL is applicable not only to I(0) and I(1) variables but also to variables of mixed order. However, this method cannot be applied if the variables are integrated of order two or more. The bound test is applied to test the presence of a long-run relationship among the variables. It assumes that the variables are I(0) and I(1) and results in lower bound and upper bound critical values. The calculated lower bound critical values show that all variables are I(0), while the upper bound critical value implies that the variables are I(1). The calculated F-statistic must be larger than the upper bound critical value for a specified significance level to reject null of no cointegrating relationship. Null of no long-run relationship cannot be rejected if the calculated F-statistic is smaller than the lower bound critical values. The test is inconclusive if its estimate falls between lower and upper bound critical values for a specific significance level.

## 4. Empirical Analysis

In this section, the econometric methods discussed in Section 3.2 are employed on real data to check whether the correlation between the return of the Indonesian stock market is constant or time varying. We also use the autoregressive distributed lag method to estimate the parameters in Equation (3) and to identify which factors can be used to explain the dynamic conditional correlation.

### 4.1. Descriptive Statistics

Figure 1 shows the historical evolution of the uncertainty of economic policy in the developed markets and the return of the Indonesian stock market. It is evident from the figure that stock market returns are more volatile than global policy uncertainty. In addition, an increasing trend is evident in global policy uncertainty post the terrorist attack on 9/11 2001; during the Iraq war in 2003, the global financial crisis in 2008–2009, and the 2011 debt ceiling; and in regard to the concerns about the Chinese economy in late 2015

and the Brexit referendum in June 2016 [84]. The return of the Indonesian stock market also reflects a declining trend during the same period.

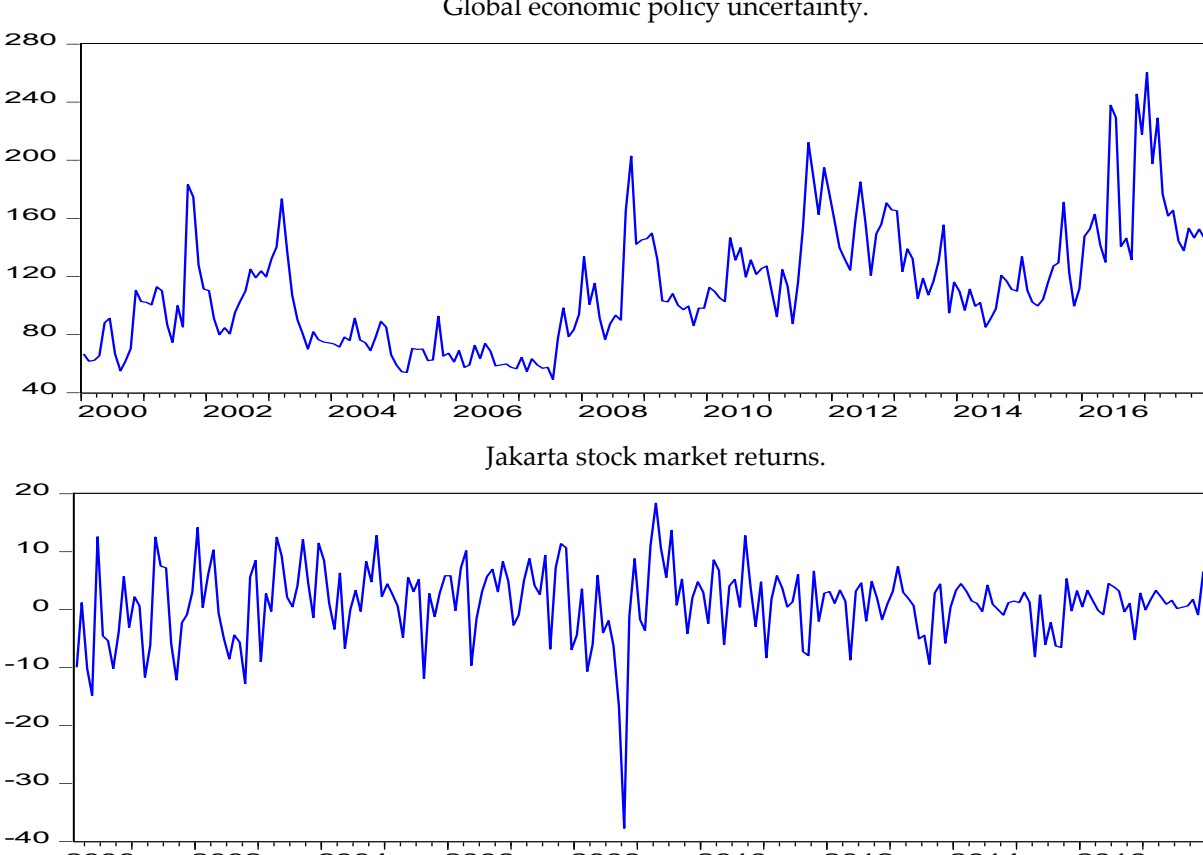

**Figure 1.** Evolution of global EPU and the return of the Indonesian stock market.

Table 1 shows the descriptive statistics of our data series. Both variables show large variability. Kurtosis estimates are larger than those of the normal distribution, implying the non-normal distribution of the variables. Jarque-Berra test statistics further confirm this finding. Before examining the dynamic conditional correlation, it is important to check the stationarity and heteroscedasticity of the time series [41]. The augmented Dickey–Fuller (ADF) test with a constant and the Dickey–Fuller (DF) test with a structural break confirm the stationarity of the data series in level. The ARCH (LM) test statistics also reject the null of homoscedasticity. Finally, the unconditional correlation between economic policy uncertainty and stock market returns is negative.

**Table 1.** Descriptive statistics of EPU and Jakarta stock market returns.

|  | **JKSE** | **GEPU** |
| --- | --- | --- |
| Mean | 1.070 | 111.970 |
| Std | 6.498 | 41.324 |
| Skewness | −1.122 | 0.916 |
| Kurtosis | 8.188 | 3.857 |
| JB | 286.250 * | 36.783 * |
| Unconditional correlation between | −0.147 ** |  |
| JKSE and GEPU | (−2.170) |  |
| ADF (Constant) | −11.689 * | −3.947 ** |
| DF (Structural Break) | 13.536 * | 5.720 * |
| ARCH (2) LM Test | 3.391 ** | 4.532 ** |

Note: Std refers to the standard deviation; * and ** indicate significance at 1% and 5%, respectively. The estimated break dates for Jakarta stock returns and global economic policy uncertainty are October 2008 and July 2007, respectively.

*4.2. Rolling Window Correlation*

In the next step, we calculate the rolling window correlation between the uncertainty of economic policy in the developed markets and the return of the Indonesian stock market. The fixed window of fifty observations is used to estimate the rolling correlation. Table 2 illustrates that on average, there is a negative correlation between the uncertainty of economic policy in the developed markets and the return of the Indonesian stock market. On the basis of the Jarque-Berra normality test, the null of the normal distribution of rolling window correlation is rejected. Hence, we conclude that the rolling correlation between policy uncertainty and the return of the Indonesian stock market is time varying. This is also demonstrated in Figure 2. The figure indicates that the rolling correlation remained highly volatile during the sample period. The sign of the correlation also changes over time.

**Table 2.** Descriptive statistics of rolling window correlation.

|  | **Mean** | **Std** | **Skewness** | **Kurtosis** | **JB** |
|---|---|---|---|---|---|
| Rolling Correlation | −0.1633 | 0.1997 | −0.0253 | 1.8647 | 9.9869 |

Note: Std and JB refer to standard deviation and Jarque Berra, respectively.

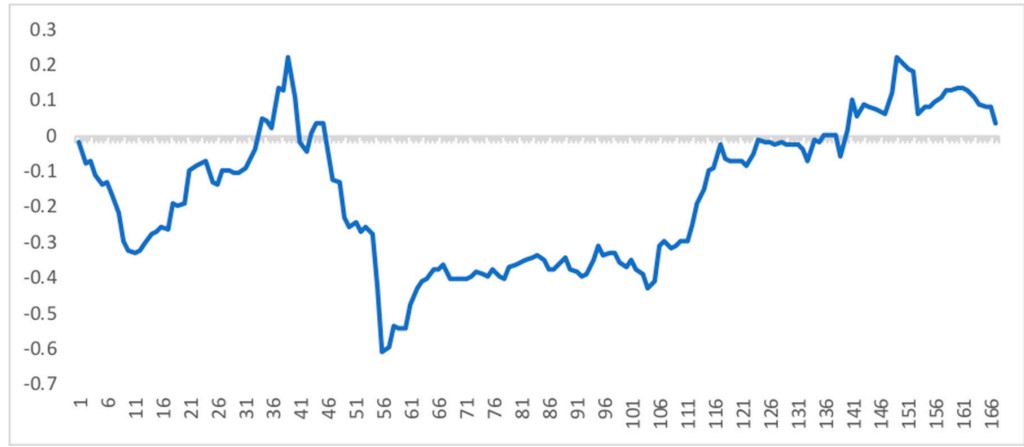

**Figure 2.** Rolling window correlation between GEPU and JKSE.

*4.3. Dynamic Conditional Correlation Results*

Table 3 provides descriptive statistics of dynamic conditional correlation. The average estimate of conditional correlation is negative. Hence, our results are consistent with those of prior studies conducted by [40,41,62] The negative estimate of the average conditional correlation may imply that when facing liquidity constraints in one country, investors may sell off their assets in another country to raise funds either because these funds are needed today or in the future [85]. It also shows that the prices of stocks in the Indonesian stock market will decrease when global policy uncertainty increases. Negative skewness indicates the non-symmetric distribution of the conditional correlation. Higher than normal estimates of Kurtosis statistics show that the series under consideration has non-normal distribution. This finding is further confirmed by the [86] test statistics that strongly reject the null of a normal distribution of the conditional correlation.

Table 4 shows the estimates of variance ($\lambda_{is}$) and covariance ($\delta$) decay factors. It is apparent from the table that the decay factors are highly significant and close to unity. The estimated degrees of freedom for the *t* distribution are significant and well below 30, which confirm that the *t*-distribution is appropriate for capturing the fat-tailed nature of the distribution of asset returns [81].

**Table 3.** Descriptive statistics of the dynamic conditional correlation (DCC).

| Variable | Mean | Std | Skewness | Kurtosis | JB |
|---|---|---|---|---|---|
| DCC | −0.168328 | 0.1154 | −0.567098 | 2.530553 | 12.30539 |

Note: Std and JB represent standard deviation and Jarque Berra normality test, respectively.

**Table 4.** Maximum likelihood estimates of the t—DCC model converged after 26 iterations.

| Parameter | Estimate | Standard Error | T-Ratio [Prob] |
|---|---|---|---|
| $\lambda_1$ | 0.92448 | 0.025413 | 36.3783[0.00] |
| $\lambda_2$ | 0.98549 | 0.0095123 | 103.6013[0.00] |
| $\Delta$ | 0.97543 | 0.025228 | 38.6644[0.00] |
| Dof | 5.3871 | 1.3321 | 4.0440[0.00] |

Note: dof refers to degrees of freedom.

Figure 3 presents the evolution of the conditional correlation. It indicates that the correlation remained highly volatile during the sample period. Apart from 2001M9, 2001M10, 2006M2, 2006M3, and 2006M4, the dynamic conditional correlation for the rest of the study period was negative. However, for the period beginning from 2007M4 to 2010M11, the dynamic conditional correlation became more negative. This is the global financial crisis period that resulted in an increase in global economic policy uncertainty and led to a larger decrease in Indonesian stock market returns. Based on Figure 3 and the [86] test statistics, we conclude that the conditional correlation between the uncertainty of economic policy in the developed markets and the return of the Indonesian stock market is time varying.

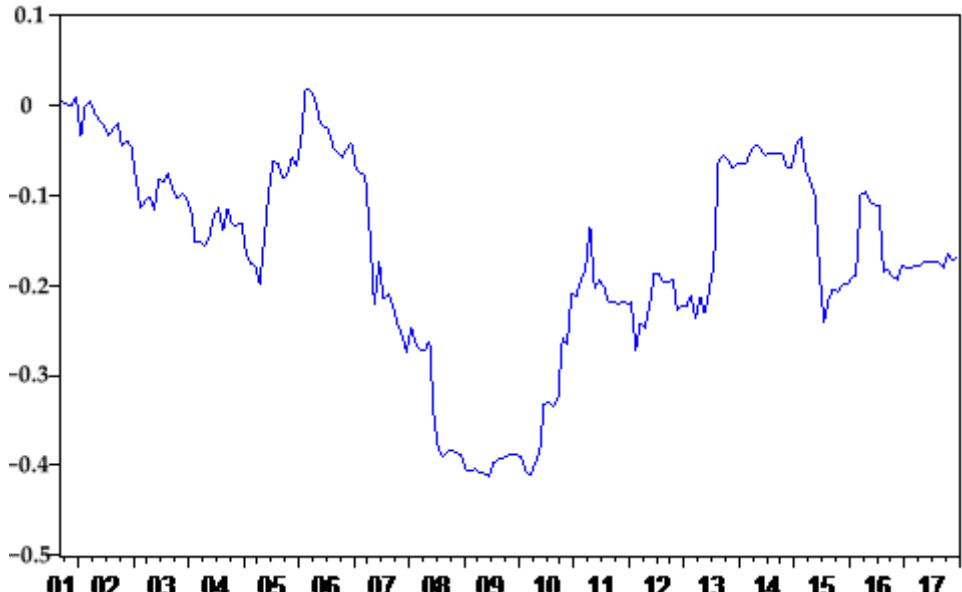

**Figure 3.** DCC between global EPU and Jakarta stock market returns.

### 4.4. Autoregressive Distributed Lag Model

The ARDL approach discussed in Section 3.2.3 is applied here to simultaneously examine the long-run and short dynamics of Equation (3). Although ARDL does not require testing the time series properties of the variables, it is important to confirm that none of the variables integrates an order of two or more. To test the time-series properties of the variable before estimating the model, the augmented Dickey–Fuller (ADF) test is applied. The ADF test results given in Table 5 show that all variables are I(1) and I(0) in log level and log first difference, respectively. The Dickey–Fuller test with structural breaks further confirms this finding for all variables, except inflation, which is stationary in level and first difference.

**Table 5.** Unit root tests with intercept specification.

| Variable | Augmented Dickey−Fuller Test | | Dickey–Fuller Test with Structural Break | |
|---|---|---|---|---|
| | Level | 1st Difference | Level | 1st Difference |
| $cpi_t$ | −0.1424 | −11.350 * | −5.787 * | −14.823 * |
| $gcop_t$ | −2.279 | −10.611 * | −4.356 | −11.196 * |
| $gdp_t$ | −2.797 | −3.019 | −4.013 | −21.352 * |
| $wcop_t$ | −2.871 | −12.537 * | −3.969 | −14.169 * |
| 1% Critical values | −4.006 | −4.006 | −5.347 | −5.347 |

Note: * denotes one 1% significance level.

Equation (4) is estimated in three different forms. Model 1 includes a recessionary dummy for the entire sample period. We isolate the effect of the recession dummy into individual recessionary periods and re-estimate the equation (Models 2 and 3). The bound cointegration test results given in Table 6 show that the calculated F-statistic is larger than the upper bound critical values for Models 1 and 3 at a significance level of one percent. For Model 2, the calculated F-statistic is larger than the upper bound critical value at a significance level of five percent. Hence, the null of no long-run relationship among the variables is rejected at the one percent significance level for Models one and three and at the five percent significance level for Model two.

**Table 6.** Bounds cointegrationtest—ARDL (2, 8, 0, 3, 0).

| | Model 1 | | Model 2 | | Model 3 | |
|---|---|---|---|---|---|---|
| Calculated F Statistic | 4.548 | | 4.008 | | 4.629 | |
| Significance Level | Pesaran et al. (2001) [87] | | Pesaran et al. (2001) [87] | | Pesaran et al. (2001) [87] | |
| | LB | UB | LB | UB | LB | UB |
| 1 percent | 3.29 | 4.37 | 3.29 | 4.37 | 3.29 | 4.37 |
| 5 percent | 2.56 | 3.49 | 2.56 | 3.49 | 2.56 | 3.49 |
| 10 percent | 2.2 | 3.09 | 2.2 | 3.09 | 2.2 | 3.09 |

Note: LB and UB represent lower bound and upper bound respectively.

After establishing the presence of cointegration, the next step is to simultaneously examine the long- and short-run dynamics of Equation (3). Long-run estimates of the estimated models are given in Table 7. It is apparent from the table that all of the estimated parameters are significant at a five percent significance level. Further, they confirm their theoretical prediction. An increase in inflation, a rise in global crude oil prices, and an increase in world crude oil production increase the vulnerability of the equity returns to the uncertainty of economic policy shocks of developed markets. The real income estimate is negative and significant. This means that an increase in real income will reduce the vulnerability of the country's stock market returns to the uncertainty of economic policy shock of developed markets and, hence, will have a negative effect on the dynamic conditional correlation.

**Table 7.** Long-run estimates of ARDL model (2, 8, 0, 3, 0).

| Variable | Model 1 | Model 2 | Model 3 |
|---|---|---|---|
| Constant | 15.344 (2.281) ** | 17.528(2.558) ** | 17.479(2.572) ** |
| $cpi_t$ | 1.862(4.094) ** | 2.075(3.821) ** | 2.051(3.794) ** |
| $gcop_t$ | 0.30(6.293) ** | 0.311(6.341) ** | 0.306(6.329) ** |
| $gdp_t$ | −0.892(−4.185) ** | −0.941(−4.126) ** | −0.935(−4.010) ** |
| $wcop_t$ | 1.691(3.571) ** | 1.461(2.530) ** | 1.450(2.720) ** |

Note: ** denotes that the estimated parameters are at a five percent significance level.

Table 8 shows the short-run coefficients of the estimated models. The lagged estimate of the dynamic condition correlation is positive and significant, which indicates persistence in the conditional correlation between the uncertainty of economic policy in the developed markets and the return of the Indonesian stock market. The inflation effect on the conditional correlation is negative at most of the lags in the short run and contradicts its long-run estimate. The real income effect is negative in the contemporary period and at lag two. However, at lag one, the real income estimate is positive, which is not in conformity with its long-run estimate.

**Table 8.** Error correction model (2, 8, 0, 3, 0).

| Variable | Model 1 | Model 2 | Model 3 |
|---|---|---|---|
| $\Delta dcc_{t-1}$ | 0.189(2.73) ** | 0.167(2.353) ** | 0.169(2.421) ** |
| $\Delta cpi_t$ | −0.829(−2.489) ** | −0.761(2.297) ** | −0.779(−2.359) ** |
| $\Delta cpi_{t-1}$ | 0.380(0.938) | 0.406(1.004) | 0.392(0.974) |
| $\Delta cpi_{t-2}$ | −0.830(−2.360) ** | −0.801(−2.282) ** | −0.807(−2.303) ** |
| $\Delta cpi_{t-3}$ | −0.259(−0.984) | −0.242(−0.920) | −0.248(−0.948) |
| $\Delta cpi_{t-4}$ | −0.048(−0.186) | −0.026(−0.10) | −0.030(−0.117) |
| $\Delta cpi_{t-5}$ | −0.666(−2.755) ** | −0.632(−2.617) ** | −0.638(−2.650) ** |
| $\Delta cpi_{t-6}$ | 0.018(0.073) | 0.041(0.169) | 0.035(0.144) |
| $\Delta cpi_{t-7}$ | −0.730(−3.118) ** | −0.685(−2.920) ** | −0.696(−2.971) ** |
| $\Delta gdp_t$ | −0.442(−2.720) ** | −0.423(−2.611) ** | −0.432(−2.682) ** |
| $\Delta gdp_{t-1}$ | 0.412(2.132) ** | 0.420(2.179) ** | 0.416(2.165) ** |
| $\Delta gdp_{t-2}$ | −0.281(−1.685) | −0.272(−1.641) | −0.273(−1.647) |
| $RI$ | −0.003(−2.166) ** | | |
| $RI_1$ | | 0.002(0.518) ** | |
| $RI_2$ | | −0.008(−1.891) | −0.006(−2.055) ** |
| $RI3$ | | −0.004(−2.459) ** | −0.004(−2.494) ** |
| $ect_{t-1}$ | −0.115(−5.300) ** | −0.114(−4.977) ** | −0.115(−5.348) ** |
| $R^2$ | 0.22 | 0.23 | 0.23 |
| DW | 2.03 | 2.02 | 2.04 |
| F statistic LM test | 0.69[0.69] | 0.552[0.7] | 0.62[0.65] |
| F statistic ARCH test | 3.331[0.012] | 3.15[0.02] | 3.098[0.02] |
| F statistic Ramsey RESET test | 0.809[0.6] | 1.091[0.372] | 1.032[0.414] |

Note: ** denotes that the estimated parameters are at a five percent significance level. t-values and probability values are given in parentheses and square brackets, respectively; DW, LM, and ARCH refer to Durbin–Watson, Lagrange multiplier, and autoregressive conditional heteroscedasticity, respectively.

The effect of the overall recession on the dynamic conditional correlation is negative and significant. We also isolate the effect of the recession dummy into individual recessionary periods. The estimated results of Model 2 show that apart from RI3, none of the recessionary indicators has a significant effect on the conditional correlation. However, RI2 and RI3 appear to have a significant negative effect when RI1 is dropped and the model is re-estimated (Model 3). The coefficient of the error correction term shows the speed of adjustment of the dependent variable towards its long-run equilibrium level after a shock. A large estimate of error correction term indicates speedy adjustment. Our estimate of error correction term ranges between 0.114 and 0.115. This means that 11.4 to 11.5 percent of the disequilibrium resulting from the shock in the previous month is corrected back to the long-run equilibrium in the current month.

The residual diagnostic test statistic results indicate that the estimated model is well specified and has no serial correlation issue. However, there is a heteroscedasticity issue with residuals of the estimated models. This results in under- and over-estimation of standard error, which leads to incorrect estimation of t-values of the estimated parameters. To address the issue, standard errors of the estimated parameters are adjusted with the Newey–West heteroscedasticity test.

A cumulative test of recursive residuals is applied to test the stability of the estimated model. Parameter instability is conceived if the calculated sum of recursive residuals

stands beyond five percent critical bounds. According to the information in Figure 4, it is confirmed that the cumulative sum remains within critical bounds. Hence, our findings conclude that the short- and long-run estimates of the selected models are stable and that there is no structural break during the sample period.

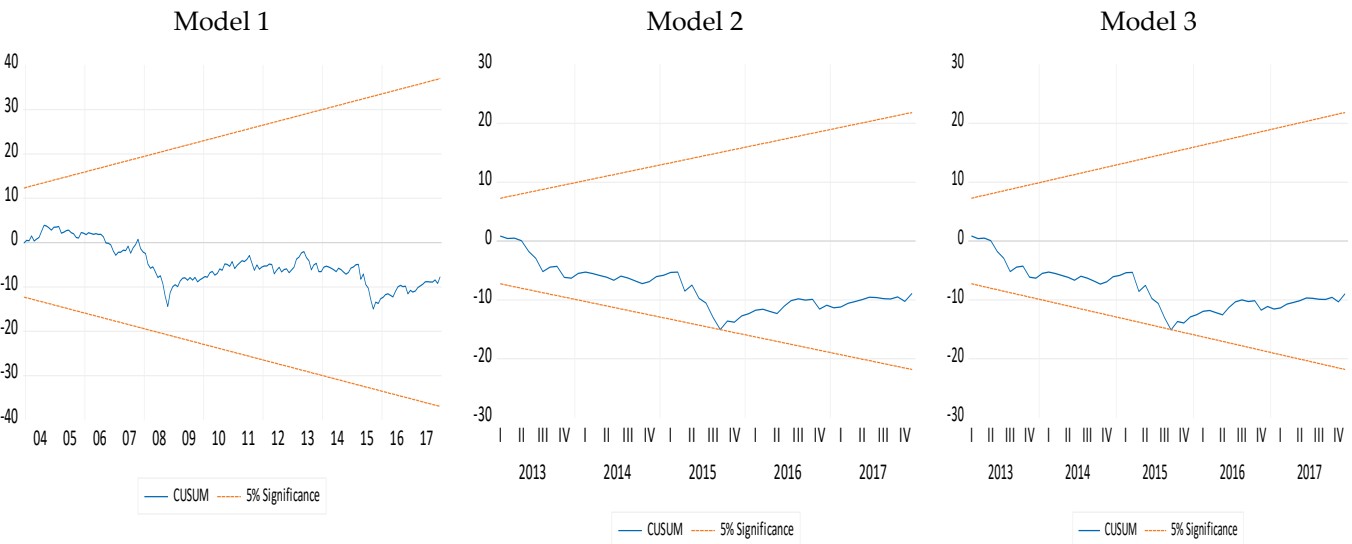

**Figure 4.** Cumulative sum of recursive residuals.

## 5. Conclusions

Trade liberalization and financial integration in the emerging markets are at the heart of policy making and research. As is well known, a rise in international trade and financial integration could increase risk in policy uncertainty. Shocks originating from the world-leading economies are inclined to affect emerging markets. As a result, the global stock market tumbles, and the global economy is exposed to vulnerability. To understand this phenomenon, several studies have examined the relationships between trade and financial integration of emerging markets. As an illustration, ref. [88] examined the stock market volatility of financial markets in emerging and developed countries. Likewise, ref. [89] conducted a study on the level of stock market integration for twenty-five emerging economies.

There is no dearth of literature on policy uncertainty and stock returns. However, to date no particular study has examined the relationship between trade and financial integration for the Indonesian stock market. Thus, to bridge the gap, our study provides new evidence on the relationship between economic policy uncertainty and the stock market return of Indonesia.

In this paper, we focused on the dynamic interaction between the uncertainty of economic policy in developed markets and the return of the Indonesian stock market. The objective was to identify the nature of the correlation between the uncertainty of economic policy in the developed markets and the return of the Indonesian stock market and the factors that explain this correlation.

Our study offers a sound framework that covers both risk and uncertainty to test the effect of uncertainty of economic policy with reference to the Indonesian economy. From the perspective of investors, it is significant to comprehend the direction and nature of the correlation between the uncertainty of economic policy in the developed markets and the return of the Indonesian stock market.

In order to examine the effect of global economic policy on the return of the Indonesian stock market, we first tested whether or not the correlation between the policy uncertainty in the global economy and the return of the Indonesian stock market is constant. Afterward, we determined to what degree oil price shocks, macroeconomic variables, and recessionary

indicators affect the conditional correlation between the variables. Furthermore, we examined whether our recessionary indicators, inflation, global crude oil prices, gross domestic product, and world crude oil production, are appropriate in explaining the dynamic conditional correlation between policy uncertainty in the global economy and the return of the Indonesian stock market.

This paper employed both rolling window correlation and dynamic conditional correlation to test the first hypothesis and determine the integration, also known as co-movement, synchronization, and the correlation between the financial markets of the world-leading economies and Indonesia. It was observed that shocks hitting the global economy eventually spread to other countries, including Indonesia.

Our study presents some interesting facts. It is revealed the correlation between global policy uncertainty and the return of the Indonesian stock market is time varying with positive and negative values for separate periods. The average negative estimate of time-varying correlation indicates that when confronted with liquidity constraints in one country, investors may sell off their assets in another country and raise funds in order to meet current or future financial needs. The resulting capital outflow arising from the negative effect of policy uncertainty in the developed markets on the stock market returns also has a negative effect on the sustainable growth of Indonesia.

Several policy implications can be drawn from our research. Our empirical results suggest that the Indonesian stock market is not profitable for global investors, particularly when the global policy uncertainty is high. Regression results based on the autoregressive distributed lag method show that global crude oil prices, world crude oil production, inflation, and gross domestic product have a significant effect on the conditional correlation. Considering the above observations, it is recommended that Indonesian policymakers should closely monitor global crude oil prices and its production, inflation, and growth prospects for avoiding vulnerability of the stock market returns to global uncertainty shocks.

Like most of the studies, our research also has some limitations. First, it examines the interaction between the uncertainty of economic policy in developed markets and the return of the Indonesian stock market; second, global crude oil prices, world crude oil production, gross domestic product, consumer price index, and recessionary indicators are used as determinants of the dynamic conditional correlation.

There is plenty of room for further research. Future studies might be focused on finding out whether the correlation between the world uncertainty index for Indonesia and the return of the Indonesian stock market is constant or time-varying. This can be further extended by splitting aggregate oil price shocks into supply-side shocks, aggregate demand shocks, and oil-specific demand shocks. Such studies will be very instrumental in understanding the shocks that potentially have a significant effect on the dynamic conditional correlation between the uncertainty of economic policy in developed markets and the return of the Indonesian stock market.

**Author Contributions:** All authors contributed to the study's conceptualization and design. The conceptual ideas and theoretical framework were developed by S.M.H. The methodology and literature review were performed by M.A.G., while data analysis and result interpretation were carried out by W.-K.W. All authors have read and agreed to the published version of the manuscript.

**Funding:** This research is funded by Yancheng Institute of Technology, Project Number XJR 2019066, and Asia University, China Medical University, Hang Seng University of Hong Kong, and the Ministry of Science and Technology (MOST) (Project Numbers 106-2410-H-468-002 and 107-2410-H-468-002-MY3).

**Institutional Review Board Statement:** Not applicable.

**Informed Consent Statement:** Not applicable.

**Data Availability Statement:** Data is contained within the article.

**Acknowledgments:** The authors would like to thank the Editor, Editor Assistant for their guidance and follow-up. Also, authors would like to thank the anonymous referees for their valuable comments to enhance our paper to appear in its current format. The third author would like to thank Robert B. Miller and Howard Thompson for their continuous guidance and encouragement.

**Conflicts of Interest:** The authors hereby declare no conflict of interest.

**Ethical Approval:** This article does not contain any studies with human participants or animals performed by any of the authors.

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
