# Peer review of "Sustainability of Global Economic Policy and Stock Market Returns in Indonesia"

_sustainability, doi:10.3390/su13105422_

Round 1

Reviewer 1 Report

The approach used in this paper is interesting, but there are some considerations:

  • The math equations are illegible
  • The tables are chaotic-in table 5 * means a 1% significance level, and in table 7 - 5% significance level
  • there is no information about the assumptions imposed on the parameters in the DCC-GARCH model and there is no verification of them at the stage of estimation
  • The RI1 period is written differently (pp. 6)
  • No description of table 3 (there are no such parameters in the formulas)
  • There is no clear explanation of what models 1, 2, 3 are why there are different periods in the figures
  • Where is the first and second hypothesis in the text?It is only in the conclusions
  • Additional letters and numbers appear in the text (e.g. title of table 6)
  • In the bibliographic list no paper from Sustainability Why do Authors want to publish an article in this journal?

Author Response

Thank you very much for your valuable comments and feedback, which have helped us to improve this manuscript significantly, making it appropriate for your readership.

We would also like to send our appreciation to you for your time and efforts in reviewing our paper. We would like to thank you for your following comments:

  • Is the content succinctly described and contextualized with respect to previous and present theoretical background and empirical research (if applicable) on the topic? (yes)
  • Are the arguments and discussion of findings coherent, balanced and compelling? (yes)
  • Is the article adequately referenced? (yes)
  • The approach used in this paper is interesting,

We would also like to send our appreciation to you for your time and efforts in reviewing our paper and for providing excellent comments. Below are our responses to your helpful comments and suggestions.

Question 1. Are the research design, questions, hypotheses and methods clearly stated? (must be improved)

Answer 1:  Thanks for your advice. We have stated the research design, questions, hypotheses, and methods clearly in our revised manuscript.

Question 2. For empirical research, are the results clearly presented? (must be improved)

Answer 2:  Thanks for your advice. We have presented the empirical results clearly in our revised manuscript.

Question 3. The math equations are illegible.

Answer 3:  Thanks for your advice. We have improved the presentation of all the math equations, re-written all the math equations by using equation writer, and do our best to make them as much legible as possible in our revised manuscript.

Question 4. The tables are chaotic-in table 5 * means a 1% significance level, and in table 7 - 5% significance level

Answer 5:  Thanks for your advice. We have improved all the tables and corrected all mistakes so that all tables are not chaotic in our revised manuscript.

Question 6. there is no information about the assumptions imposed on the parameters in the DCC-GARCH model and there is no verification of them at the stage of estimation

Answer 6:  Thank you very much for your advice. We have stated the assumptions of the DCC-GARCH model and verified it at the stage of estimation in our revised manuscript.

Question 7: The RI1 period is written differently (pp. 6)

Answer 7:  Thank you very much for your advice. the RI1 period is correctly written and is available on page 6. This can be verified from the data sheet on Indonesian recessionary indicators downloaded from Federal Reserve Bank of St. Louis.    

Question 8: No description of table 3 (there are no such parameters in the formulas)

Answer 8:  Thank you very much for your advice. We have described Table 3 properly in our revised manuscript.

Question 9: There is no clear explanation of what models 1, 2, 3 are why there are different periods in the figures

Answer 9:  Thank you very much for your advice. We have explained the results of Models 1, 2, 3 clearly and explained why there are different periods in the figures in our revised manuscript.

Question 10: Where is the first and second hypothesis in the text?It is only in the conclusions

Answer 10:  Thank you very much for your advice. The first and second hypotheses are provided on page number 3 and highlighted red in our revised manuscript.

Question 11: The Additional letters and numbers appear in the text (e.g. title of table 6)

Answer 11:  Thank you very much for pointing out our mistakes. We have corrected the mistakes in our revised manuscript.

Question 12: In the bibliographic list no paper from Sustainability Why do Authors want to publish an article in this journal?

Answer 12:  Thank you very much for your advice. We have cited some related papers in Sustainability in our revised manuscript. We want to publish our paper in the special issue of “Behavioral Business and Behavioral Financial Economics with application” because our paper fits into the scope of the special issue.

We hope that you will find this manuscript suitable to be included in an upcoming issue of your publication.

Reviewer 2 Report

This paper investigates the effects of global economic policy uncertainty on the return of the Indonesian stock market. Exploring the time-varying correlation via both the rolling window method and time-varying built dynamic conditional correlation method, results indicate that the correlation between global policy uncertainty and Indonesian stock returns is time-varying. The findings have meaningful policy implications. The paper is well written. Empirical analysis is properly designed, executed and interpretated.

Author Response

Thank you very much for your valuable comments and feedback, which have helped us to improve this manuscript significantly, making it appropriate for your readership.

We would also like to send our appreciation to you for your time and efforts in reviewing our paper. We would like to thank you for your following comments:

  • English language and style are fine/minor spell check required (yes)
  • Is the content succinctly described and contextualized with respect to previous and present theoretical background and empirical research (if applicable) on the topic? (yes)
  • Are the research design, questions, hypotheses and methods clearly stated? (yes)
  • Are the arguments and discussion of findings coherent, balanced and compelling? (yes)
  • For empirical research, are the results clearly presented? (yes)
  • Is the article adequately referenced? (yes)
  • The findings have meaningful policy implications.
  • The paper is well written.
  • The empirical analysis is properly designed, executed and interpreted.

,

We would also like to send our appreciation to you for your time and efforts in reviewing our paper and for providing excellent comments. Below are our responses to your helpful comments and suggestions.

Question 1. Are the conclusions thoroughly supported by the results presented in the article or referenced in secondary literature? (must be improved)

Answer 1:  Thanks for your advice. We have improved the conclusion so that it is thoroughly supported by the results presented in the article or referenced in secondary literature in our revised manuscript.

We hope that you will find this manuscript suitable to be included in an upcoming issue of your publication.

Reviewer 3 Report

Comments on "Sustainability of Global Economic Policy and Stock Market Returns in Indonesia”.

Content comments:

Subject of the article is suitable for publication in Sustainability, the article provides sufficiently original and new contribution to its theme and the arguments, results and conclusions appear to be adequately justified.

The aim of this paper is to focus: i) “on the dynamic interaction between the uncertainty of economic policy in the developed markets and the return of the Indonesian stock market”; ii) “to find out the nature of correlation between the uncertainty of economic policy in the developed markets and the return of the Indonesian stock market”; iii) and to find out “the factors that explain this correlation” (lines. 551-555).

However, the discussion of the statistical results and the conclusions are poor and need to be reworked. The literature review is used in an instrumental way and does not bring much to the reader. The literature review must be put in resonance with the results of the article. There must be a dialogue between the results put forward by the authors of the article and the literature review. This is a pity because the author provides a statistically valid study. It is imperative to take the results of this article and put them in dialogue with the observations/results made by the authors/studies mentioned in the literature review.

In the conclusion, if the author wants to “enable the relevant policymakers to set up appropriate policy measures to reduce vulnerability” (lines 530-531), it might be interesting to interview some representative investors of the players in the Indonesia stock market. It might be interesting, if not useful, to bring a more qualitative analysis to the conclusions made by the authors. Beyond the statistical elements (interesting), what is the interpretation given by the different actors of this sector? If the author does not have access to people working in this sector, there are many articles, essays and analyses on this subject. It would be relevant for the quality of the article to refer to them. This would add value to the article.

Form comments:

The referencing still has some typos. Sometimes we find “et al. when more than two authors, sometimes only when more than three authors. Examples: “Arslanturk, Balcilar, and Ozdemir, 2011” (line 265) / “Arslanturk, et al 2011” (line 281). The referencing style used in the article must be rigorous and consistent.

Line 122: It is Ahmad and Sharma (2018) and not “Ahmed and Sharma (2018)”.

Line 693: The reference “Liu, L., and Zhang, T. (2015) Economic policy uncertainty and stock market volatility. Finance Research Letters, 693 Vol. 15, pp. 99-105”. is missing in the text.

Line 324: “Improvement of Johansen cointegration test (1991)”. This reference is not found in the bibliography at the end of the article.

This reference is not used rigorously in the text: “Jacque, C.M., and Berra, A.K. (1980) Efficient tests for normality, homoscedasticity and serial independence of regression residuals. Economic Letters, Vol. 6(3), pp. 255-259” (line 668). For examples: lines 353; 373; 381; 397; 402; 406.

The style used in the bibliography is not always consistent.

Author Response

Thank you very much for your valuable comments and feedback, which have helped us to improve this manuscript significantly, making it appropriate for your readership.

We would also like to send our appreciation to you for your time and efforts in reviewing our paper. We would like to thank you for your following comments:

  • English language and style are fine/minor spell check required (yes)
  • Is the content succinctly described and contextualized with respect to previous and present theoretical background and empirical research (if applicable) on the topic? (yes)
  • Are the research design, questions, hypotheses and methods clearly stated? (yes)
  • For empirical research, are the results clearly presented? (yes)

  • Subject of the article is suitable for publication in Sustainability,

  • the article provides sufficiently original and new contribution to its theme and

  • the arguments, results and conclusions appear to be adequately justified.

  • the author provides a statistically valid study.

We would also like to send our appreciation to you for your time and efforts in reviewing our paper and for providing excellent comments. Below are our responses to your helpful comments and suggestions.

Question 1. Are the arguments and discussion of findings coherent, balanced and compelling?        (must be improved)

Answer 1:  Thank you very much for your advice. We have improved the arguments and discussion of findings coherent, balanced and compelling in our revised manuscript.

Question 2. Is the article adequately referenced? (can be improved)

Answer 2:  Thank you very much for your advice. We have improved the references so that our manuscript is adequately referenced in our revised manuscript.

Question 3. Are the conclusions thoroughly supported by the results presented in the article or referenced in secondary literature? (must be improved)

Answer 3:  Thanks for your advice. We have improved the conclusion so that it is thoroughly supported by the results presented in the article or referenced in secondary literature in our revised manuscript.

Question 4. However, the discussion of the statistical results and the conclusions are poor and need to be reworked.

Answer 4:  Thanks for your advice. We have improved the discussion of the statistical results and the conclusions in our revised manuscript.

Question 5. The literature review is used in an instrumental way and does not bring much to the reader. The literature review must be put in resonance with the results of the article.

Answer 5:  Thank you very much for your advice. We have improved the review of the literature to put in resonance with the results of the article in our revised manuscript.

Question 6. There must be a dialogue between the results put forward by the authors of the article and the literature review.

Answer 6: Thank you very much for your advice. We have improved the review of the literature so that it is a dialogue between the results of our paper and the literature review in our revised manuscript.

Question 7. This is a pity because the author provides a statistically valid study. It is imperative to take the results of this article and put them in dialogue with the observations/results made by the authors/studies mentioned in the literature review.

Answer 7:  

Thank you very much for your advice. We have improved the review of the literature to make it imperative to take the results of our paper and put them in dialogue with our observations/results mentioned in the literature review in our revised manuscript.

Question 8. In the conclusion, if the author wants to “enable the relevant policymakers to set up appropriate policy measures to reduce vulnerability” (lines 530-531), it might be interesting to interview some representative investors of the players in the Indonesia stock market. It might be interesting, if not useful, to bring a more qualitative analysis to the conclusions made by the authors. Beyond the statistical elements (interesting), what is the interpretation given by the different actors of this sector? If the author does not have access to people working in this sector, there are many articles, essays and analyses on this subject. It would be relevant for the quality of the article to refer to them. This would add value to the article.

Answer 8:  Thank you very much for your advice. Kindly note that our research is purely empirical analysis and publicly available data. Thus, hence conducting interviews is beyond the scope of our paper. However, to address your comments, we have thoroughly improved the literature section with new citations highlighted in yellow. We sincerely hope you are satisfied with our revision in this matter in our revised manuscript.

Question 9. The referencing still has some typos. Sometimes we find “et al. when more than two authors, sometimes only when more than three authors. Examples: “Arslanturk, Balcilar, and Ozdemir, 2011” (line 265) / “Arslanturk, et al 2011” (line 281). The referencing style used in the article must be rigorous and consistent.

Answer 9:  Thank you very much for your advice. We have made our citations consistent in our revised manuscript.

Question 10. Line 122: It is Ahmad and Sharma (2018) and not “Ahmed and Sharma (2018)”.

Answer 10:  Thank you very much for your advice. Xxx in our revised manuscript.

Question 11. Line 693: The reference “Liu, L., and Zhang, T. (2015) Economic policy uncertainty and stock market volatility. Finance Research Letters, 693 Vol. 15, pp. 99-105”. is missing in the text.

Answer 11:  Thank you very much for pointing out our mistake. We have removed it from the bibliography of our revised manuscript since we have not cited it in our manuscript, so we have dropped it from the bibliography.

Question 12. Line 324: “Improvement of Johansen cointegration test (1991)”. This reference is not found in the bibliography at the end of the article.

Answer 12:  Thank you very much for pointing out our mistake. We have included it in the bibliography of our revised manuscript.

Question 13. This reference is not used rigorously in the text: “Jacque, C.M., and Berra, A.K. (1980) Efficient tests for normality, homoscedasticity and serial independence of regression residuals. Economic Letters, Vol. 6(3), pp. 255-259” (line 668). For examples: lines 353; 373; 381; 397; 402; 406.

Answer 13:  Thank you very much for your advice and information. We have cited this paper and included it in the reference to our revised manuscript.

Question 14. The style used in the bibliography is not always consistent.

Answer 14:  Thank you very much for pointing out our problem. We have made the bibliography consistent in our revised manuscript.

We hope that you will find this manuscript suitable to be included in an upcoming issue of your publication.
